# Water Environment Characteristics and Water Quality Assessment of Water Source of Diversion System of Project from Hanjiang to Weihe River

**DOI:** 10.3390/ijerph20042890

**Published:** 2023-02-07

**Authors:** Wei Wu, Hang Chen, Sheng Xu, Ting Liu, Hao Wang, Gaoqing Li, Jiawei Wang

**Affiliations:** 1State Key Laboratory of Eco-Hydraulics in Northwest Arid Region, Xi’an University of Technology, Xi’an 710048, China; 2Shaanxi Han Weihe Water Diversion Engineering Construction Co., Ltd., Xi’an 710086, China

**Keywords:** water source, comprehensive evaluation of water quality, hierarchical clustering analysis, spatial and temporal distribution, influence factors

## Abstract

The water source of the water diversion project from the Hanjiang River to the Weihe River is one of the most important drinking water sources in China. Its water quality is related to the water safety of the long-distance water diversion system from the Hanjiang to Weihe Rivers. In order to explore the spatiotemporal change trend of the water environment characteristics of the water source area and analyze the key factors that have a greater impact on it, this study collected 9 types of water environment physical and chemical parameters from 10 water quality monitoring sections from 2017 to 2019; the water environment characteristics of the water source area of the water diversion system from the Hanjiang River to the Weihe River were analyzed and evaluated by using the variance analysis method, the hierarchical cluster analysis method and the water quality identification index evaluation method. The results were as follows. (1) There was spatiotemporal heterogeneity in a number of physical and chemical parameters in the water body of the water source. In terms of time, the concentrations of COD_Mn_, COD, BOD_5_ and F^−^ were higher in the flood season (July–October) than in the non-flood season (November–June). The concentrations of DO, TP and TN in the non-flood season were higher than those in the flood season. Spatially, the concentration of physical and chemical parameters of the water body in the Huangjinxia Reservoir area was higher than that in the Sanhekou Reservoir area. (2) The water quality of the water source area was good. The comprehensive water quality reached the Class II water quality standard of surface water environmental quality. Time showed that the comprehensive water quality in the non-flood season was better than that in the flood season. Spatially, the overall water quality of the tributaries was better than that of the mainstream. TN is a key indicator that affects water quality. (3) The spatial and temporal differences in water quality in water source areas are mainly affected by factors such as rainfall, temperature and human activities. This study can provide a scientific and data basis for related research on maintaining and improving the quality of the ecological environment of the water source areas of the Hanjiang to Weihe River Water Diversion System.

## 1. Introduction

With the growth of the population and economic development, water resources have become a bottleneck restricting human survival and sustainable development [1]. The uneven distribution of water resources and unbalanced water demands of human society make it necessary to transfer water across river basins. Cross-basin water transfer is not only one of the most important ways to solve the problem of the inconsistent allocation of water resources, land, labor and other resources, but also an important strategic task to ensure the sustainable and coordinated development of the national society, economy and environment [2,3]. In the whole water diversion system, the water source area, water conveyance area and water receiving area together constitute a complex water environment system, and accidents in any link of the system will affect the water quality safety of the whole system. Once the water quality fails to meet the standard, not only will it seriously affect people’s normal lives, but it will also cause huge economic losses and adverse social and political impacts [4]. Therefore, it is very important to study the water environment characteristics and water quality evaluation of the water source areas of water diversion systems to ensure the continuous normal operation of water diversion projects and ensure the quality and safety of the water supply [5].

At present, the main evaluation methods used at home and abroad are the single-factor evaluation method, pollution index method, fuzzy comprehensive evaluation method, identification index method, principal component analysis method and neural network method [6,7]. The newly developed water pollution index method has been applied to the analysis of water quality for groundwater samples by Son et al. [8], which is flexible and easy to calculate. The artificial neural network and the improved fuzzy clustering technology were used by S Azimi et al. for potential drinking water quality assessment in a GIS environment [9]. The result shows that the water quality of most aquifers in Iran has a significant decline trend. The multivariate statistical analysis method was used by Ustao ğ Lu and Tepe to analyze the spatial and temporal distribution characteristics of water quality factors, and to evaluate water quality using the identification index method [10]. The water diversion project from the Hanjiang River to the Weihe River is a long-distance water diversion project, which connects the two major water systems of the Hanjiang River and the Weihe River in the north and south. The project has two water source areas, the Huangjinxia Reservoir and the Sanhekou Reservoir, which are located on the main stream of the Hanjiang River and its tributary Ziwu River, respectively. The water is diverted from the Huangjinxia Reservoir through the pump station, and flows to the control gate of the Sanhekou Reservoir from the “Huangsan Section Tunnel” by gravity. At the same time, the water from the Sanhekou Reservoir is gathered. The water is diverted from the “Crossing Section Tunnel” by gravity to “Huangchigou”, and connected to the water supply network in the Guanzhong region [11]. For such a complex inter-basin water transfer system, to ensure the realization of the water environment planning objectives of the water transfer project, maximize the project benefits and ensure the safe operation of the water transfer project, it is necessary to more systematically study the space–time change characteristics and main influencing factors of the water quality of the water source area of the Han-Wei River Diversion Project [12,13].

This study takes the water source area of the Hanjiang to Weihe River Water Diversion Project as the research object, and evaluates the water quality of the water source area and identifies the main pollution factors by means of multivariate statistical analysis based on the water quality monitoring data of each section from 2017 to 2019 and the data of on-site sampling and detection. Significant difference analysis on the time and space scale of water quality in the project water source area is carried out by analysis of variance (ANOVA). The water quality data are grouped by hierarchical cluster analysis (HCA) in time and space to further analyze the spatio-temporal distribution characteristics of water quality in water source areas. Based on the grouping of cluster analysis, the water quality of each group and its data are evaluated by the water quality identification index method, and the main pollution factors are identified. The purpose is to provide a guarantee for the operational safety of the water diversion project from Hanji to Weihe River, and to provide ideas for the prevention and control of water environment pollution, water quantity and quality dispatching and water environment planning and the management of water diversion projects across river basins.

## 2. Materials and Methods

### 2.1. General Situation of the Study Area

The Hanjiang to Weihe River Water Diversion Project connects the two water systems of the Hanjiang River and Weihe River. The two water sources are the Huangjinxia Reservoir and Sanhekou Reservoir, respectively. The main tributaries of the Huangjinxia Reservoir Area are the Jinshui River, Youshui River, Liangxin River, Dangshui River, etc. The main tributaries of the Three Estuaries Reservoir Area are the Jiaoxi River, Pu River and Wenshui River (Figure 1). The total reservoir capacity of the Huangjinxia Reservoir Area and Sanhekou Reservoir Area is 229 and 710 million m^3^, respectively. The regulated reservoir capacity is 269 and 660 million m^3^, respectively. They are a daily reservoir and multi-year reservoir, respectively, and the normal water level and dead water level are 450, 440 and 643, 558 m, respectively. The Huangjinxia Reservoir Project Area and Sanhekou Reservoir Project Area belong to a subtropical climate and the humid and semi-humid climate in the north subtropical zone, respectively. The annual average air temperature, extreme maximum air temperature and minimum air temperature in the Huangjinxia Reservoir Area and Sanhekou Reservoir Area are 14.5, 39.4, −11.9 and 12.3, 37.4, −16.4 °C, respectively. The annual average precipitation, annual average wind speed and annual average evaporation are 806 mm, 1.2 m/s, 1065.6 mm and 901 mm, 1.2 m/s, 1213 mm, respectively [14].

### 2.2. Data Sources

The basic data for the study are the monitoring data from 2017 to 2019 of the Yangxian Environmental Protection Bureau. We selected the pH, dissolved oxygen (DO), permanganate index (COD_Mn_), chemical oxygen demand (COD), five-day biochemical oxygen demand (BOD_5_), fluoride (F^−^), ammonia nitrogen (NH_3_-N), total phosphorus (TP) and total nitrogen (TN) of 10 monitoring sections, and evaluated the water quality according to the Environmental Quality Standards for Surface Water [15]. The flow data are from the monitoring data of Yangxian Station on the Han River. See Figure 1 for monitoring sections.

### 2.3. Analysis Method

#### 2.3.1. ANOVA

Analysis of variance (ANOVA), also known as the “F test”, was proposed by Professor Fisher [16], and it can test the significance of the difference between the mean values of two or more samples [17,18]. According to the number of control variables, it is divided into single-factor and multi-factor ANOVA. In this study, univariate multiple variance analysis is used. The control variable is time or space (section), and its calculation formula is as follows:(1)$${\mathrm{SS}}_{t}=\overset{k}{\underset{j=1}{\sum }}\overset{n}{\underset{i=1}{\sum }}X_{\mathrm{ij}}^{2}-\frac{{\left(\sum_{j=1}^{k}\sum_{i=1}^{n}x_{\mathrm{ij}}\right)}^{2}}{\mathrm{nk}}$$
(2)$${\mathrm{SS}}_{W}=\overset{k}{\underset{j=1}{\sum }}\overset{n}{\underset{i=1}{\sum }}X_{\mathrm{ij}}^{2}-\frac{\sum_{j=1}^{k}{\left(\sum_{i=1}^{n}x_{\mathrm{ij}}\right)}^{2}}{n}$$
(3)$${\mathrm{SS}}_{b}={\mathrm{SS}}_{t}-{\mathrm{SS}}_{W}$$
(4)$${\mathrm{MS}}_{W}=\frac{{\mathrm{SS}}_{W}}{k\left(n-1\right)}$$
(5)$${\mathrm{MS}}_{b}=\frac{{\mathrm{SS}}_{b}}{n-1}$$
where SS_t_ is the total variation, SS_w_ is intra-group variation, SS_b_ is inter-group variation, K is the number of data groups, N is the number of data in the j-th group, X_ij_ is the i data in group j, MS_w_ is the intra-group variance, MS_b_ is the variance between groups, n − 1 is the degree of freedom within the group, and k(n − 1) is the degree of freedom between groups.

F is greater than Fα [(n − 1), k(n − 1)] (α = 0.05), indicating that there are significant differences in the mean values of each population at different levels under the control variable, and the calculation formula is
(6)$$F=\frac{{\mathrm{MS}}_{b}}{{\mathrm{MS}}_{W}}=\frac{{\mathrm{kSS}}_{b}}{{\mathrm{SS}}_{w}}$$

#### 2.3.2. Cluster Analysis

Cluster analysis, also known as group analysis [11,12], is a multivariate statistical analysis method that classifies samples or indexes based on the principle of clustering. The basic idea is to group objects with similar properties according to the degree of affinity between samples or indicators. The hierarchical clustering analysis method is used in the study. The Euclidean distance square method is used to calculate the distance between samples, and the difference square method, also called the Ward method, is used to calculate the distance between groups.

#### 2.3.3. Analysis of Comprehensive Water Quality Indicator

The comprehensive water quality identification index is composed of integer digits and three or four decimal digits [13,19]. Its structure is
(7)$$I_{\mathrm{wq}}=X_{1}\;\times \;X_{2}X_{3}X_{4}$$
where X_1_ is the comprehensive water quality category of the river; X_2_ is the position of comprehensive water quality in the range of water quality change of type X_1_, thus realizing the comparison of water quality in the same type of water; X_3_ is the number of individual indexes that are inferior to the target of the water environment functional area in the water quality evaluation; X_4_ is the comparison result of the comprehensive water quality category and water body functional area category, and its value depends on the pollution degree of comprehensive water quality.

The formula for calculating X_1_ × X_2_ is as follows:(8)$$X_{1}\;\times \;X_{2}=\frac{1}{m}\sum \left(P_{1}+P_{2}+\ldots +P_{m}\right)$$
where m is the number of individual water quality indexes participating in the comprehensive water quality assessment, and Pm is the single-factor water quality index of the m-th water quality factor.

X_3_ is the number of individual indexes that are inferior to the target of the water environment functional area in the water quality indexes participating in the evaluation (i.e., X_3_ = 0, indicating that all water quality indexes participating in the evaluation have met the target of the water environment functional area; X_3_ = 1, indicating that one of the indexes participating in the comprehensive water quality evaluation has not met the target of the functional area, and so on).

X_4_ serves to judge whether the comprehensive water quality category is inferior to the functional area category of the water environment. If the comprehensive water quality category is better than or reaches the functional area category, X_4_ = 0; if the water quality category is inferior to that of the water environment functional area and X_2_ is not 0 in the comprehensive water quality identification index, then X_4_ = X_1_ − f, and f is the water environment functional area category in the formula; if the water quality category is inferior to that of the water environment functional area and X_2_ in the comprehensive water quality identification index is 0, then X_4_ = X_1_ − f − 1.

### 2.4. Data Analysis

The concentration diagram of physical and chemical indexes was drawn using Origin2018. ANOVA, cluster analysis and correlation analysis were performed in IBM SPSS Statistics 26 and Rstudio 4.2.1 software.

## 3. Results and Discussion

### 3.1. Analysis of Spatio-Temporal Distribution Characteristics

The time distribution characteristics in Figure 2 show that the pH, dissolved oxygen (DO), permanganate index (COD_Mn_), chemical oxygen demand (COD), five-day biochemical oxygen demand (BOD_5_), fluoride (F^−^), ammonia nitrogen (NH_3_-N), total phosphorus (TP), total nitrogen (TN) and total nitrogen (TN) take values of 6.59–8.82, 6.0–11.95, 1.0–3.9, 4.0–14.0–0, 0.5–3.27, 0.05–0.65, 0.03~0.49, 0.005~0.09, 0.078~3.4 mg/L, respectively. The highest average pH occurred in December at 7.80, and the lowest in July at 7.43. The highest DO and TP mean values were found in February, with the mean values of 8.27 and 0.04 mg/L, respectively, and the lowest averages in June and November were 7.29 and 0.02 mg/L, respectively. The mean values of COD_Mn_, BOD_5_, F^−^ and TN were the highest in August, October, July and March, with the mean values of 2.76, 2.16, 0.22 and 1.14 mg/L, respectively, and the lowest in December, with the mean values of 1.84, 1.56, 0.14 and 0.72 mg/L, respectively. The highest mean values of COD and NH_3_-N occurred in October and July, respectively, with the mean values of 11.05 and 0.23 mg/L, and the lowest in January, with the mean values of 8.21 and 0.17 mg/L, respectively. The variation in COD_Mn_, COD, BOD_5_ and F^−^ in the year was similar, which shows that the concentration in the flood season is higher than that in the non-flood season. TN and TP are affected by human activities and have high concentrations in the non-flood season. With the exception of TN, the water quality indexes all meet the Class II water quality standard for surface water environmental quality.

The spatial distribution characteristics in Figure 3 show that the highest mean values of pH, COD and BOD_5_ occur in the Youshui River, the Yangcheng Section of the Hanjiang River and the Yishui River Section, respectively, with the mean values of 7.89, 10.68 and 2.39 mg/L. The highest mean values of DO, TP and TN were found in the section of Huangjinxia of the Hanjiang River, with the mean values of 8.39, 0.05 and 2.29 mg/L, respectively. The highest mean values of COD_Mn_, F^−^ and NH_3_-N were found in the Dangshui River Section, with the mean values of 2.70, 0.27 and 0.33 mg/L, respectively. The lowest mean values of pH, DO and TN were found in the Jinshui River, Dangshui River and Youshui River, respectively, with the mean values of 7.14, 7.27 and 0.40 mg/L. The lowest mean values of COD_Mn_, COD, BOD_5_ and TP were found in the upstream section of the Jiaoxi River in Foping County, with the mean values of 1.72, 8.31, 1.24 and 0.01 mg/L, respectively. The lowest mean values of F^−^ and NH_3_-N were found in the Xujiacheng section of the Wenshui River, with the mean value of 0.12 mg/L.

The water quality concentrations of COD_Mn_, COD, BOD_5_, F^−^ and NH_3_-N in the sections of the Wenshui River and Jiaoxi River in the Sanhekou Reservoir Area were lower than those in the Golden Gorge Reservoir Area, and the water quality of the Sanhekou Reservoir Area was better than that of the Golden Gorge Reservoir Area. The average concentration of COD_Mn_, BOD_5_, F^−^ and NH_3_-N in the Dangshui River is the highest, and its water quality is relatively poor. As the Dangshui River is located in the city of Yangxian County, it is strongly disturbed by human activities. The TN concentration in the main stream of the Hanjiang River is relatively high and it fails to meet the Class II water quality standard for surface water environmental quality.

Based on the spatio-temporal variation characteristics of different water quality parameters in the above water source areas, the spatio-temporal distribution differences of different water quality physical and chemical parameters are further clarified by using one-factor ANOVA (Table 1). The results show that the water quality indexes of the water sources have significant differences in different sections and times. Among them, pH, DO, COD_Mn_, BOD_5_, F^−^, NH_3_-N, TP and TN have significant differences in different sections. The mean concentrations of COD_Mn_, BOD_5_, F^−^, NH_3_-N in the Dangshui River are the highest, and their water quality is relatively poor. The TN concentration in the trunk of the Hanjiang River is higher than that in all tributaries. COD_Mn_, COD, BOD_5_, F^−^ and NH_3_-N in all sections of the Sanhekou Reservoir Area are better than those in the Huangjinxia Reservoir Area. DO, COD_Mn_, COD, BOD_5_ and TN have significant differences in different months, in which DO tends to increase with the decrease in temperature in winter, while COD_Mn_, COD and BOD_5_ change in the same year, which shows that the concentration in the flood season is higher than that in the non-flood season.

### 3.2. Analysis of Correlation and Clusters

#### 3.2.1. Correlation Analysis

The Pearson correlation coefficients (Table 2) of the water quality indices of 10 water quality monitoring sections from 2017 to 2019 in the water source area of the Hanjiang to Weihe River Water Diversion Project are calculated, and the correlations of the water quality indices are analyzed to clarify the background relationship of each index before it is transformed by the comprehensive water quality indicator method.

The correlation coefficient of the water quality index in Table 2 shows that DO is significantly negatively correlated with NH_3_-N, and DO is significantly negatively correlated with BOD_5_ and F^−^. COD_Mn_ is positively correlated with COD and BOD_5_. COD is positively correlated with BOD_5_. NH_3_-N is positively correlated with TN. There is no correlation among other physical and chemical parameters of water quality.

#### 3.2.2. Cluster Analysis

In order to further analyze the spatial and temporal distribution patterns of water quality in water sources, this study uses hierarchical cluster analysis to cluster water quality indicators in time and space (Figure 4 and Figure 5). In terms of time, the sequence is divided into the non-flood season (T1: November~June of the next year) and the flood season (T2: July~October). In space, the sequence is divided into the tributaries of the Han River (S1: 1—Wenshui River, 2—Jiaoxi River, upper reaches of Foping County, 3—Jiaoxi River, lower reaches of Foping County, 6—Dangshui River, 7—Youshui River, 8—Jinshui River, 9—Huangjin Gorge) and the main stream of the Han River (S2: 4—Chengyang section of the Han River, 5—Yishui River, 10—Yangxi section of the Han River). Figure 4 and Figure 5 show that the water quality of the water source area has significant differences in time (flood season, non-flood season) and space (trunk and tributary), but there are no significant differences in water quality conditions in each month within the group.

### 3.3. Analysis of Comprehensive Assessment of Water Quality

According to the definition of the single-factor water quality indicator and comprehensive water quality indicator, the mean values of the water quality indicators in each group are statistically analyzed. The results of the water quality assessment are shown in Table 3 and Table 4.

The evaluation results of the single-factor water quality identification index method show that TN fails to meet the Class II water quality standard for surface water environmental quality, and other physical and chemical parameters meet the Class II water quality standard for surface water environmental quality, in which DO, COD_Mn_, NH_3_-N and TP meet the Class II water quality standard and COD, BOD_5_ and F^−^ meet the Class I water quality standard. The evaluation results of the comprehensive water quality identification index method show that the comprehensive water quality of T1 is classified as Class I, the water quality of T2 is worse than that of T1, and the comprehensive water quality is classified as Class II. The comprehensive water quality of group S1 and S2 reaches Class I, but according to the comprehensive water quality identification index, the water quality of group S1 is better than that of group S2. The water quality of the Dangshui River and Huangjinxia Reservoir is relatively poor, and the comprehensive water quality category is II. The comprehensive water quality of other sections meets the water quality standard for surface water environmental quality category I. TN is a parameter exceeding the standard water quality index, and TN exceeds the standard in each section in different periods. As a whole, the TN load in the water source area exceeds the target water quality requirement for a long time and a wide range, and it is the key index affecting the overall water quality.

## 4. Discussion

### 4.1. Driving Factors of Temporal and Spatial Changes in Water Quality Characteristics

Temporally, the changes in COD_Mn_, COD, BOD_5_ and F^−^ in this study are similar in the year, which shows that the concentration in the flood season (July–October) is higher than that in the non-flood season (November–June). This may be because the sewage treatment in Foping County, Yangxian County, Hanzhong County, Chenggu County, Xixiang County and other counties where the water source is located fails to meet the standards, and the guidelines on rain and sewage separation have not been implemented, which results in the mixing of sewage and rainwater during the flood season, and the sewage volume increases rapidly [20]. In addition, the rainfall in the flood season is higher than that in the non-flood season, which makes the input of external pollutants in water in agriculture, plant and soil higher, resulting in high water physical and chemical indexes [20]. The DO concentration is higher in the non-flood season than in the flood season. In general, DO varies with the season. In winter and spring, the air temperature is low and the dissolved oxygen in the water is high, while, in summer, the air temperature is high and the dissolved oxygen in the water is low [21]. The concentration of DO in the non-flood season is higher than that in the flood season due to the change in water temperature caused by seasonal variation. The concentration of TN and TP in the non-flood season is higher than that in the flood season. A possible reason is that there is more agriculture and planting in the Huangjinxia Reservoir Area and Sanhekou Reservoir Area [20]. Nitrogen and phosphorus in fertilizers, medicines and the soil of plants flow into the river with irrigation and rainfall in spring and autumn. In addition, the decrease in flow rate during the non-flood season increases the concentration of TN and TP [22,23]. High flow during the flood season will affect the dilution and buffering of pollutants [24].

Spatially, the concentrations of COD_Mn_, COD, BOD_5_, F^−^ and NH_3_-N in the sections of the Wenshui River and Jiaoxi River in the Sanhekou Reservoir Area are lower than those in the sections of the Huangjinxia Reservoir Area, and the water quality of the Sanhekou Reservoir Area is better than that of the Huangjinxia Reservoir Area. The main reason may be that the number of cities and towns in the area of the Huangjinxia Reservoir is greater than that of the Sanhekou Reservoir, and its land area, population density and industrial and agricultural scale are larger than those of the Sanhekou Reservoir [25,26,27,28]. As a result, the human sewage and industrial and agricultural pollution inflow into the area of the Huangjinxia Reservoir is higher than that in the Sanhekou Reservoir.

### 4.2. Analysis of Water Quality Characteristic Factors

Correlation analysis of the physical and chemical parameters of water shows that there is a significant positive correlation between COD_Mn_ and COD and BOD_5_. The reason is that both of them indirectly and relatively represent the important water quality index of the organic substance quantity in water samples by using the principle of oxidizing organic substances [29], and studies show that there is a good positive correlation between COD_Mn_ and COD and BOD_5_. DO is negatively correlated with BOD_5_ and F^−^ significantly [16,30,31], which may be due to the higher water temperature in summer, which accelerates the production and metabolism of aquatic organisms and the ion exchange frequency between water and sediment, resulting in a decrease in DO content and increase in BOD_5_ and F^−^ in the water body [32]. The decrease in the water temperature in autumn slows down the production and metabolism of aquatic organisms and the frequency of ion exchange between water and sediment, which results in an increase in DO content and decrease in BOD_5_ and F^−^ in the water body [33]. NH_3_-N is significantly negatively correlated with DO and positively correlated with TN. The reason is that TN refers to the total nitrogen content in various forms of inorganic nitrogen (nitrate nitrogen, nitrite nitrogen, ammonia nitrogen) and organic nitrogen in water [34,35,36]. When large quantities of domestic sewage, farmland wastewater or industrial wastewater containing nitrogen are discharged into the water, the content of organic and inorganic nitrogen compounds in the water and the total nitrogen content increase, and biological and microbial species reproduce in large quantities, so that the dissolved oxygen in the water is increased [37]. This leads to a decrease in dissolved oxygen content and the deterioration of the water quality [37]. Ammonia nitrogen refers to nitrogen in the form of free ammonia and ammonium salt in water [36]. The composition of molecular ammonia and ionic ammonium is affected by the temperature. The lower the water temperature, the smaller the proportion of molecular ammonia in water [26], and the more the DO is affected by the water temperature. The higher the water temperature, the larger the proportion of molecular ammonia [26] and the lower the DO. The results show that there is a correlation between ammonia nitrogen and total nitrogen at a certain proportion [36,38,39].

Hierarchical clustering analysis was carried out on the water quality index of the water source area in time and space. It was found that the water quality index was clustered into two periods in time: T1 (non-flood season): November-June and T2 (flood season): July-October. There are two types of spatial clustering: S1 (tributary of Hanjiang River): 1—Wenshui River, 2—Jiaoxi River upstream of Foping County, 3—Jiaoxi River downstream of Foping County, 6—Dangshui River, 7—Youshui River, 8—Jinshui River and 9—Huangjinxia Reservoir, and S2 (trunk of Hanjiang River): 4—Hanjiang Chengyang Section, 5—Yishui River, 10—Hanjiang Yangxi Section. This is basically similar to the clustering in related studies [33,40,41]. In Section 3.1, the annual variation basically shows the distribution rules of COD_Mn_, COD, BOD_5_ and F^−^ in the flood season being higher than in the non-flood season and DO, TN and TP being higher in the non-flood season than in the flood season, which indicates the rationality of clustering water quality indexes into the flood season and non-flood season in time. At the same time, the water quality of the Sanhekou Reservoir Area is better than that of the Huangjinxia Reservoir Area in space, while the Sanhekou Reservoir Area and Huangjinxia Reservoir Area basically correspond to the tributaries of Lele clustering into two categories: the trunk of the Hanjiang River and the branch of the Hanjiang River.

### 4.3. Law and Cause of Water Quality Comprehensive Evaluation Index

The result of the single-factor water quality identification index method shows that the TN water quality factor meets the water quality standard of Class II in super water functional areas and other water quality factors meet the requirements of water functional areas. The reasons may be the unreasonable use of nitrogen fertilizers in agriculture, the incorrect discharge of pollutants in industry and human activities and water and soil erosion. Relevant research shows that long-term and unreasonable nitrogen fertilizer input in agriculture will bring a large nitrogen load to the soil environment and may pose a great threat to the surface water [33]. Li et al. indicated that rural sewage and livestock and poultry culture discharges were also important factors causing nitrogen pollution in the upper reaches of the Hanjiang River in their study on the nitrogen pollution characteristics of the water body in the upper reaches of the Hanjiang River [42]. Kuo et al. mentioned in their study that nitrogen pollution in the Jinshui River Basin mainly comes from atmospheric precipitation, chemical fertilizers, soil organic nitrogen and domestic sewage [43]. Chen et al. considered that water and soil loss was the main source of nitrogen and phosphorus pollution in the Hanjiang River Basin. It is not difficult to see that human activities are one of the most important factors affecting water quality in the watershed [44].

Table 3 and Table 4 indicate that the long-term TN load and large-scale exceedance of target water quality requirements may be key indicators affecting the overall water quality. Previous analysis results show that TN is higher in the non-flood season than in the flood season, and the comprehensive water quality identification index method results show that the comprehensive water quality of T1 (non-flood season) is better than that of T2 (flood season). The reason is that the calculation of X_1_ × X_2_ in the comprehensive water quality index method is the average of each single-factor water quality index [13,19]. From the discussion in Table 1 and Section 4.1, it can be seen that most water quality indexes have significant differences in time, and most of them show that the concentration in the flood season is higher than that in the non-flood season.

According to the comprehensive water quality index, the water quality of group S1 (branch of Hanjiang River) is better than that of group S2 (trunk of Hanjiang River). The reason is that the main stream of the Hanjiang River in Yangxian County has a greater pollutant input from discharging outlets into the river [45], while group S1 located in the tributary of the Hanjiang River receives much less discharge than S2 from outlets in the river. In addition, the land area, population density and industrial and agricultural scale of Yangxian, where the trunk of the Hanjiang River is located, are larger than those of Foping County, where the branch of the Hanjiang River is located [25,26,27,28]. The human domestic sewage and industrial and agricultural pollution leading to inflow into the S2 group are higher than those of the S1 group due to human activities. The water quality of S-6 (Dangshui River) and S-9 (Huangjinxia Reservoir) is relatively poor. The reason may be that S-6, located in Yangxian County town, is closely disturbed by human activities, while S-9, located at the dam site of Huangjinxia and the construction of a water conservancy complex, may have an impact on water quality [46,47].

## 5. Conclusions

The physical and chemical parameters of the water body in the water source area have spatio-temporal heterogeneity, showing that the concentrations of COD_Mn_, COD, BOD_5_ and F^−^ are higher in the flood season than in the non-flood season, while the concentrations of DO, TP and TN are higher in the non-flood season than in the flood season. Spatially, the concentrations of the water’s physical and chemical parameters in the Huangjinxia Reservoir Area are higher than those in the Sanhekou Reservoir Area.The evaluation results of the water quality in the water source area of the Hanjiang to Weihe River Water Diversion Project by the comprehensive water quality identification index method show that the comprehensive water quality category of each section meets the water quality requirements of surface water environmental quality class II. In terms of time, the water quality of the water source in the non-flood season is better than that in the flood season. Spatially, the water quality of the branch water source is better than that of the main stream. According to the single-factor water quality index, the main pollution factor is total nitrogen.The spatio-temporal differences in water quality in water source areas are mainly controlled by factors such as rainfall, temperature and human activities, and human activities with different intensities will affect the water quality in the watershed to varying degrees.

## Figures and Tables

**Figure 1 ijerph-20-02890-f001:**
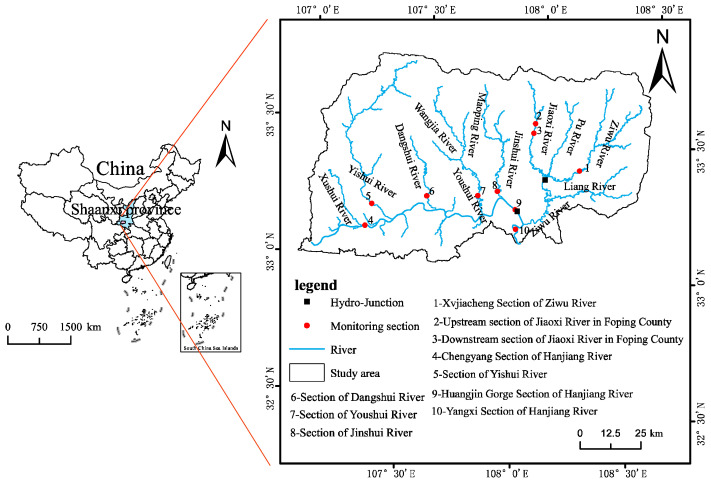
Survey of study area and distribution of water quality monitoring points.

**Figure 2 ijerph-20-02890-f002:**
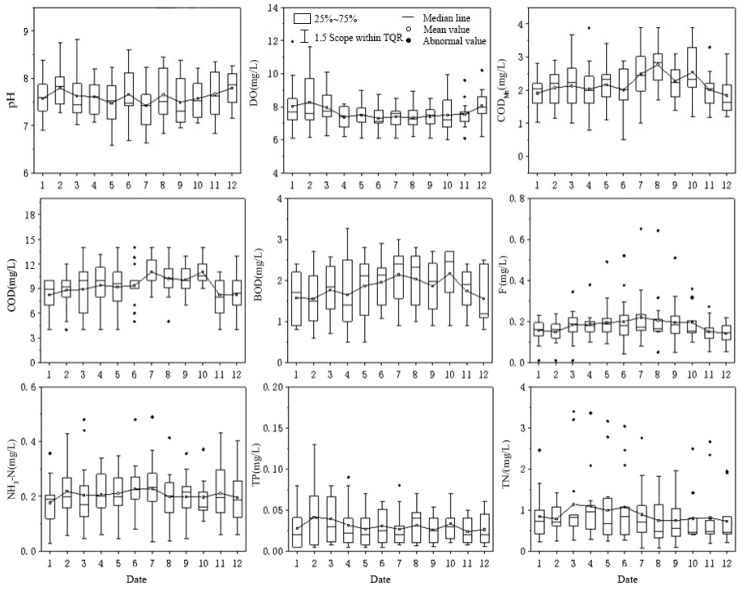
Time change of water quality in the monitoring section of the water source area during the year.

**Figure 3 ijerph-20-02890-f003:**
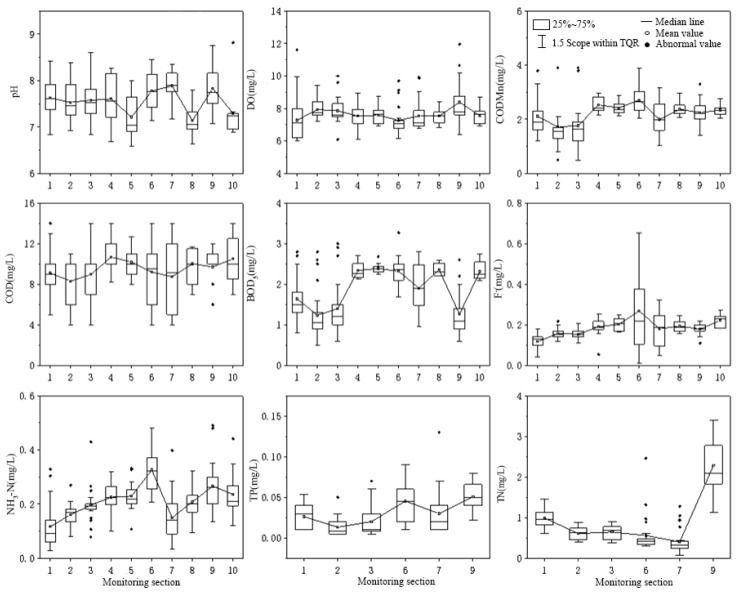
Spatial changes of water quality in monitoring sections of water sources. Note: **1**—Wenshui River Section of Xujiacheng; **2**—Jiaoxi River upstream of Foping County; **3**—Jiaoxi River downstream of Foping County; **4**—Hanjiang Chengyang Section; **5**—Yishui River Section; **6**—Dangshui River Section; **7**—Youshuihe Section; **8**—Jinshuihe Section; **9**—Hanjiang Huangjinxia Section; **10**—Hanjiang Yangxi Section.

**Figure 4 ijerph-20-02890-f004:**
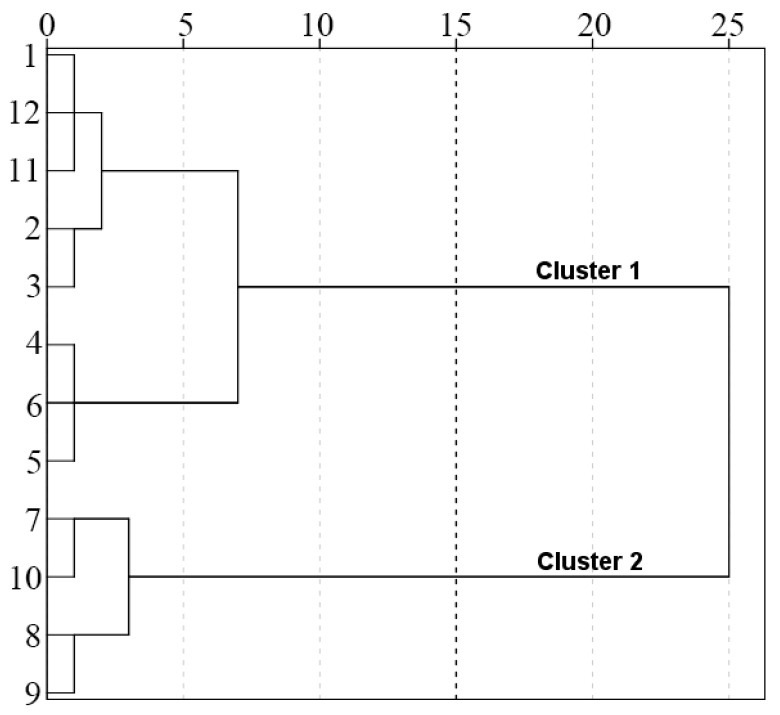
Time clustering of water quality in water source areas.

**Figure 5 ijerph-20-02890-f005:**
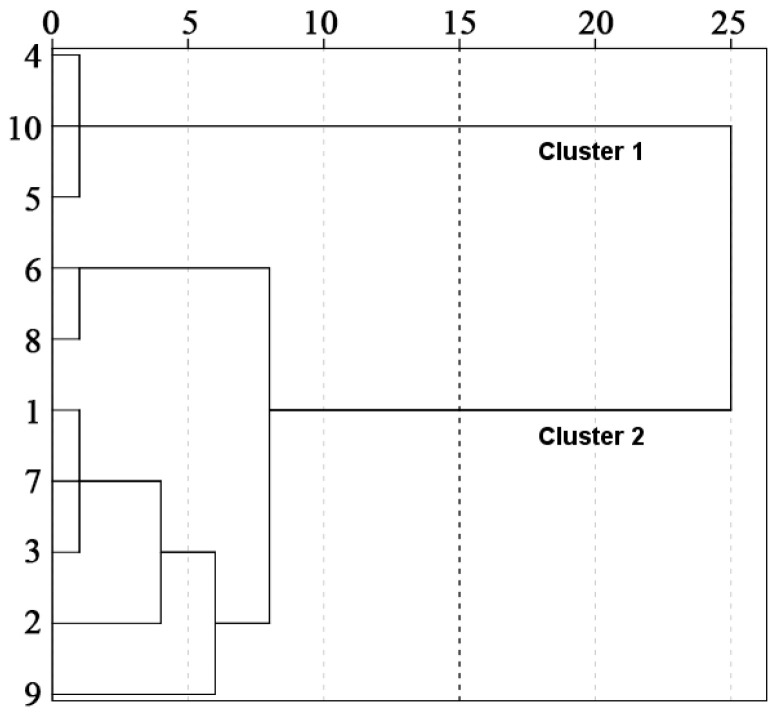
Spatial clustering of water quality in water source areas.

**Table 1 ijerph-20-02890-t001:** Time difference and spatial difference of water quality factors.

	pH	DO	COD_Mn_	COD	BOD_5_	F^−^	NH_3_-N	TP	TN
Spatial	6.22 **	2.28 *	4.79 **	1.60	16.63 **	6.20 **	15.26 **	12.36 **	95.99 **
Time	0.80	1.78 *	2.58 **	2.96 **	2.79 **	1.46	0.61	1.35	2.74 **

Note: The value is the F value; * means significant at the *p* < 0.05 level; ** means significant at the *p* < 0.01 level.

**Table 2 ijerph-20-02890-t002:** Correlation analysis of water quality factors.

Index	pH	DO	COD_MN_	COD	BOD_5_	F^−^	NH_3_-N	TP	TN
pH	1								
DO	−0.196	1							
COD_MN_	−0.175	−0.271	1						
COD	−0.104	−0.054	0.451 **	1					
BOD_5_	0.193	−0.288 *	0.723 **	0.650 **	1				
F^−^	−0.223	−0.337 *	0.073	−0.274	−0.176	1			
NH_3_-N	−0.075	−0.375 **	−0.261	−0.077	−0.274	−0.153	1		
TP	−0.001	−0.228	0.060	−0.098	−0.053	−0.255	0.278	1	
TN	0.076	−0.244	−0.193	0.224	0.024	−0.134	0.372 **	0.238	1

Note: * means significant at the *p* < 0.05 level; ** means significant at the *p* < 0.01 level.

**Table 3 ijerph-20-02890-t003:** Water quality label index evaluation table (Time).

Time	Single-Factor Water Quality Labeling Index								I_wq_	Category
	DO	COD_MN_	COD	BOD_5_	F^−^	NH_3_-N	TP	TN		
T1	1.80	2.00	1.60	1.60	1.20	2.20	2.10	3.91	2.010	Ⅰ
T2	2.10	2.30	1.70	1.70	1.20	2.20	2.10	3.61	2.110	Ⅱ

**Table 4 ijerph-20-02890-t004:** Water quality label index evaluation table (Spatial).

Spatial	Single-Factor Water Quality Labeling Index								I_wq_	Category
	DO	COD_MN_	COD	BOD_5_	F^−^	NH_3_-N	TP	TN		
S-1	2.10	2.10	1.60	1.50	1.10	1.80	2.10	4.01	2.010	Ⅰ
S-2	1.70	1.90	1.60	1.40	1.20	2.00	1.70	3.21	1.810	Ⅰ
S-3	1.80	1.90	1.60	1.50	1.20	2.10	2.00	3.31	1.910	Ⅰ
S-6	2.20	2.40	1.60	1.80	1.30	2.50	2.30	3.11	2.210	Ⅱ
S-7	2.00	2.00	1.60	1.60	1.20	2.00	2.10	2.70	1.900	Ⅰ
S-8	2.00	2.20	1.60	1.80	1.20	2.20	/	/	1.800	Ⅰ
S-9	1.40	2.10	1.60	1.40	1.20	2.30	2.40	6.14	2.310	Ⅱ
S1	1.90	2.10	1.60	1.60	1.20	2.20	/	/	1.800	Ⅰ
S-4	2.00	2.30	1.70	1.80	1.20	2.20	/	/	1.900	Ⅰ
S-5	1.90	2.20	1.70	1.80	1.20	2.20	/	/	1.800	Ⅰ
S-10	2.00	2.40	1.70	1.80	1.20	2.20	/	/	1.900	Ⅰ
S2	2.00	2.30	1.70	1.80	1.20	2.20	/	/	1.900	Ⅰ

## Data Availability

Not applicable.

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
