# Peer review of "Water Environment Characteristics and Water Quality Assessment of Water Source of Diversion System of Project from Hanjiang to Weihe River"

_ijerph, 2023, doi:10.3390/ijerph20042890_

Round 1

Reviewer 1 Report

Wei Wu et al. submitted to IJERPH an article focusing on water environment characteristics and water quality assessment of one of the most important drinking water sources in China.

The manuscript presents copious formal and substantial criticalities, briefly reported below:

- the authors carried out the study in the period 2017-2019, but disclosing the data three years later may not guarantee the reproducibility of the method and results, especially for a dynamic resource such as the aqueous matrix.

- LL 38 and 42-43: please check it, because the same thing is written. The same for LL 48-50. Similarly up to line 56, likewise for lines 60 to 72: check the redundancy problem and solve it meticulously.

- LL 72-75: “water” is written 7 times in a single sentence, please undertake to rephrase the sentence.

- L 149: R.A.? please better define it.

- what is the added value of this study? What are take-home messages?

- references must be indicated according to the IJERPH Instructions for Authors.

The manuscript deserves to be totally revised, it is not appropriate to write a scientific article describing systematically both in the introduction and in the discussions "this Author has demonstrated this, [...] this other Author said this", a reading must definitely be fed criticism and the results must be discussed and compared with other studies present in the literature.

Reading in English highlights numerous grammatical, lexical and semantic errors, consequently it is necessary a revision by an English editing service.

In compliance with the prestige of the Journal, the science and the names of the Authors, I would suggest to Wei Wu et al. to engage with greater meticulousness, improving the part of critical thinking deriving from this manuscript (not only the technical one) and to submit the work again to the Journal you will consider more appropriate. I am sorry.

Reviewer 2 Report

The paper focuses on the spatio-temporal monitoring of water quality parameters, by including the analysis of correlation and clustering, of the Hanjiang-to-Weihe River Water Diversion. The methodology is appropriate, and the conclusions as sufficiently supported by the data. However, the paper is not well written. There are too many mistakes and unclear sentences. Therefore, I recommend the paper after major revisions, according, at least,  to the detailed comments below.

General comments.

Please check / amend the quality / editing of text throughout the paper.

some specific comments.

Title

Line 11-14. Incomplete sentence.

Line 17-18. Incomplete sentence.

Line 38-75. Many sentences are repeated. This part is unreadable in this form.

Line 79, 81, 4242, 428, 434. “et al” should be “et al.”.

Line 89-91. Unclear sentence.

Line 149-150. The author surname is incorrect.

Line 149-151. Please check the appropriateness of the cited references.

Line 164. “while” should be removed.

Line 186-188. Unclear sentence.

Line 220, line 247, and table 2. “pH”, not “PH”.

Line 302-308. Incomplete / unclear sentences.

Line 319. “group and group…”.

Round 2

Reviewer 1 Report

Thank you for the answer to my previous review. The Authors have made significant changes, which improved the quality of the paper, although the "on-site" study was conducted 4 years ago (an aspect on which I continue to remain perplexed). In substance, the manuscript appears now adequately structured.

Reviewer 2 Report

The paper has been improved and can be published